# Health Experiences of African American Mothers, Wellness in the Postpartum Period and Beyond (HEAL): A Qualitative Study Applying a Critical Race Feminist Theoretical Framework

**DOI:** 10.3390/ijerph20136283

**Published:** 2023-07-03

**Authors:** S. Michelle Ogunwole, Habibat A. Oguntade, Kelly M. Bower, Lisa A. Cooper, Wendy L. Bennett

**Affiliations:** 1Division of General Internal Medicine, Department of Medicine, Johns Hopkins University School of Medicine, Baltimore, MD 21205, USA; 2Johns Hopkins Center for Health Equity, Baltimore, MD 21287, USA; 3Division of Epidemiology and Community Health, University of Minnesota School of Public Health, Minneapolis, MN 55455, USA; 4Johns Hopkins School of Nursing, Johns Hopkins University, Baltimore, MD 21205, USA; 5Department of Population, Family and Reproductive Health, Johns Hopkins University Bloomberg School of Public Health, Baltimore, MD 21205, USA

**Keywords:** maternal health, health disparities, racial disparities, postpartum care, primary care, Black women’s health, cardiometabolic risk factors, critical race feminism, intersectionality, health equity, healthcare utilization

## Abstract

The objective of this study is to explore the cultural, social, and historical factors that affect postpartum primary care utilization among Black women with cardiometabolic risk factors and to identify the needs, barriers, and facilitators that are associated with it. We conducted in-depth interviews of 18 Black women with one or more cardiometabolic complications (pre-pregnancy chronic hypertension, diabetes, obesity, preeclampsia, or gestational diabetes) within one year of delivery. We recruited women from three early home-visiting programs in Baltimore, Maryland, between May 2020 and June 2021. We used Critical Race Feminism theory and a behavioral model for healthcare utilization as an analytical lens to develop a codebook and code interview transcripts. We identified and summarized emergent patterns and themes using textual and thematic analysis. We categorized our findings into six main themes: (1) The enduring influence of structural racism, (2) personally mediated racism in healthcare and beyond, (3) sociocultural beliefs about preventative healthcare, (4) barriers to postpartum care transitions, such as education and multidisciplinary communication, (5) facilitators of postpartum care transitions, such as patient–provider relationships and continuity of care, and (6) postpartum health and healthcare needs, such as mental health and social support. Critical race feminism provides a valuable lens for exploring drivers of postpartum primary care utilization while considering the intersectional experiences of Black women.

## 1. Introduction

In the United States, Black women are 3 to 4 times more likely to experience maternal mortality than White women. They also face similar disparities in severe maternal morbidity [1,2,3,4]. These maternal health disparities are partly driven by disparities in pre-pregnancy and pregnancy-associated cardiometabolic disease [2,5,6,7]. The term “Black women” is intentionally used throughout this document to acknowledge and honor the unique experiences and historical significance associated with the identity of Black women. By using this specific term, we aim to recognize the distinct challenges, resilience, and achievements that have shaped the lives of individuals who identify as Black women. While gender-inclusive language is important, it is equally crucial to acknowledge the nuances and complexities of different identities. The postpartum period is a critical time for improving the health of mothers who have experienced medically complicated pregnancies, such as those with pre-existing chronic diseases or cardiometabolic complications during pregnancy. Referring these women to their usual primary care providers (PCPs) in the postpartum period and encouraging them to continue their care can be an effective strategy to promote cardiometabolic health, including during future pregnancies. Primary Care Provider (PCP) as defined by the National Cancer Institute is “a doctor or other licensed medical professional who manages a person’s health care over time. Primary care providers diagnose and treat a wide range of common medical conditions. They also provide preventive care, such as disease screenings and immunizations. A primary care provider may also refer a person to a specialist or coordinate care given by a specialist”. In this article, PCPs refer to non-obstetric providers who meet the above criteria [8]. This approach may also help reduce Black–White maternal health disparities [9,10].

Despite increased recognition of the importance of postpartum primary care follow-up [11,12,13,14,15], there are challenges that limit its utilization. The transition from maternity care to post-delivery well-woman and preventive care is fragmented and lacks collaborative practice strategies [16,17]. Obstetric providers and PCPs often operate in silos, resulting in missed opportunities for communication and collaboration to manage and prevent chronic diseases. Black women are again disproportionately affected, with research citing less frequent postpartum primary care follow-ups than their White counterparts [11,18]. The lack of an effective care transition may contribute to the persistence of racial disparities in maternal health outcomes and impedes progress in closing the racial health equity gap. A deeper understanding of the drivers that influence postpartum primary care utilization is needed to inform the design of interventions that improve Black women’s health. 

While prior studies have evaluated drivers of postpartum primary care utilization [18,19,20,21], these studies have not been explicitly focused on Black women, or grounded in theoretical frameworks that help to interrogate the root causes of health inequities. These studies also fail to capture an element that is critical to understanding the experiences of racialized minority groups: The role and function of racism in daily life (See Appendix A for definitions of terms: Racialization, racism, and structural racism).

Racism operates at structural and individual levels to compromise health outcomes and plays a significant role in creating and perpetuating maternal health disparities [22,23,24]. Prather and colleagues proposed a socioecological model to explain the role of racism in the reproductive health of Black women. The authors argued that at various socio-ecological levels (historical, societal, neighborhood/community, family/interpersonal, and individual), structural- and individual-level racism function to influence health [25]. 

At the structural level, racial disparities in maternal health outcomes stem from various complex factors, including limited access to healthcare throughout pregnancy, the higher burden of chronic medical conditions, social determinants of health, and unequal distribution of high-quality care. These factors are consequences of broader structural inequities, with structural racism being identified as a root cause of these disparities [26,27,28,29,30,31].

At the individual level, there is a substantial body of literature that has examined the impact of interpersonal racism and social inequities on healthcare-seeking behaviors among Black people in the US [24,31,32,33,34,35]. These topics have been raised among Black women specifically, and as it relates to prenatal and preconception care utilization [36,37,38,39]. However, to our knowledge, no studies have explored the unique experiences of Black women and, in particular, the mediating role of interlocking systems of oppression (i.e., racism, sexism, and classism), and how this may contribute to postpartum primary care utilization. To interrogate postpartum primary care utilization experiences for Black women, a racial equity, gender equity, and social justice approach that centers the voices of Black women is needed [16]. A theory that considers intersectional marginalization (e.g., gendered racism) and calls out systems that contribute to this marginalization helps to achieve both [40,41]. We use Critical Race Feminism theory [42] to explore drivers of postpartum primary care use and consider Black women’s intersectional experiences.

As such, this study is driven by the following research questions (RQ): RQ1. What are the drivers of postpartum primary care utilization for Black women, considering cultural, social, and historical contexts about health, healthcare, wellness, and identity?RQ2. What are the needs, barriers, and facilitators of postpartum primary care for Black women?

## 2. Materials and Methods

### 2.1. Theoretical Frameworks

#### 2.1.1. Critical Race Feminism

Critical race feminism is concerned with how racism functions to produce social inequities at the intersections of race, class, and gender [42,43]. The theory is an outcropping of Critical Race Theory; however, it includes an emphasis on the multiplicative disadvantage of women of color as its distinguishing feature. Critical Race Theory is a scholarly framework that focuses on the ways in which racism operates in society to create and perpetuate inequities. It seeks to understand and transform the complex relationship between race, racism, and power, with the goal of promoting social change and justice [44,45,46]. In Figure 1, the central tenets of Critical Race Feminism (which are drawn from the insights of foundational Black feminist scholars [47,48] and critical race theorists [44,49,50]), are depicted as eight ideas that form the outer circle of the figure. The inner circle of the figure highlights the central emphasis on intersectionality within this framework. 

As a framework, Critical Race Feminism allows deep exploration of the lived experiences of societal members occupying multiple intersecting identities [42,43]. More specifically, it allows for the deconstruction of popular narratives that reinforce negative stereotypes and contribute to stigmatization, and the construction of new counternarratives that “illuminate voices of women of color and the impact of intersections of their multiplicative identities on their experiences” [51]. Critical Race Feminism also enables the exploration of structural inequality and power dynamics and allows for a nuanced discussion of how motherhood intersects with other social identities. Thus, Critical Race Feminism provides an appropriate framework for examining the experiences of Black postpartum women [52,53]. It has been used in prior work to understand how identity (and unique forms of oppression due to racism, sexism, classism, etc.) may shape healthcare access and experiences and health outcomes [54,55,56].

We utilize the concepts provided by Critical Race Feminism to (1) appropriately center Black women’s lived experiences in postpartum healthcare; (2) name racism’s contextual function in participants’ lives—including the impact on postpartum primary care utilization; (3) articulate the dominant roles that racism and other forms of oppression play in the lives of Black new mothers as they seek preventative healthcare; (4) illuminate the role of structural racism (See Appendix A for definitions of terms: Racialization, racism, and structural racism) and downstream social and structural determinants of health in shaping healthcare utilization inequities for Black postpartum mothers, and (5) consider targets for intervention development around downstream consequences of racism that are not founded in deficit ideology or negative stereotypes but instead hold fidelity to Black women’s joy, strength, resilience, and ingenuity.

#### 2.1.2. Gelberg–Andersen Behavioral Model for Vulnerable Populations

Andersen and Newman’s behavioral model of health services is a widely used model for describing and understanding healthcare utilization [57,58]. The model suggests that healthcare utilization is a function of (1) a predisposition of people to use health services (i.e., predisposing factors such as age, gender, education, marital status, health belief, etc.), (2) factors that enable or impede such use (i.e., enabling factors such as income, health insurance, community resources), and (3) people’s need for care (i.e., need factors, described as both perceived and confirmed medical need) [58]. In 2000, Andersen and colleagues presented a major revision to the original model. This revision—termed the Gelberg–Andersen behavioral model for vulnerable populations (Gelberg–Andersen behavioral model)—added domains “especially relevant to understanding the health and health-seeking behavior of vulnerable populations”, including historically marginalized racial and ethnic groups, undocumented immigrants, children and adolescents, mentally ill people, chronically ill people, disabled people, elderly people, and people living with homelessness [59]. The Gelberg–Andersen behavioral model has been used most commonly when describing healthcare utilization among homeless populations, but more recently it has been used to describe healthcare utilization among Black women recently released from prison [60] and to evaluate racially and ethnically diverse pregnant women’s use of doula services [52]. 

While the updated Gelberg–Andersen behavioral model includes social determinants of health, the role of racialization, racism, and in particular, structural racism (See Appendix A for definitions of terms: Racialization, racism, and structural racism) is not well articulated in this model. Importantly the model also does not consider intersectional identities and experiences, including those that produce enabling factors and generate resources. 

### 2.2. Methods

#### 2.2.1. Conceptual Framework

We adapted the Gelberg–Andersen behavioral model [59] with the addition of principles of Critical Race Feminism (Figure 2). The most important adaptations were to (1) name racism and (2) consider intersectional experiences of Black womanhood/motherhood in the predisposing, enabling, and need factors for postpartum primary care utilization. Figure 2 shows the levels of racism across six socioecological levels [25]: Historical, structural, institutional (healthcare in this case), community, interpersonal, and individual.

Our adapted conceptual framework also informed (1) our choice of methods (qualitative one-on-one interviews were selected to “center at the margins” and provide an opportunity for counternarrative storytelling); and (2) interview guide development, including choosing topics that focused on intersectional experiences of Black motherhood and the influence of structural racism, as well as components related to healthcare utilization (see instrumentation section below).

#### 2.2.2. Study Design

Guided by our adapted conceptual framework, we limited study inclusion to postpartum women who self-identified as Black. We conducted a qualitative study using in-depth, semi-structured interviews with Black postpartum women with cardiometabolic risk factors acquired before or during pregnancy (i.e., pre-pregnancy chronic hypertension, type 2 diabetes, obesity, preeclampsia, or gestational diabetes). The Institutional Review Board at Johns Hopkins University School of Medicine approved this study.

#### 2.2.3. Study Setting

This study was conducted in Baltimore, Maryland, a predominantly Black city (62.8% Black population) with high levels of neighborhood violence and a history of discriminatory housing practices that have led to racial and economic segregation [61,62,63]. Similar to other U.S. cities, in Baltimore, structural racism becomes visible as race-based disparities in income, education, access to care, and health outcomes persist in historically redlined areas, where chronic disease is the leading cause of mortality [64]. Importantly, these disparities extend to maternal and infant health. Black maternal mortality rates are four times higher than White maternal mortality rates. Infant mortality rates among Black infants remain double that of White infants [65].

#### 2.2.4. Contemporary Context and Race Relations

This study took place during the COVID-19 pandemic and coincided with Black Lives Matter protests sparked by the police killings of George Floyd and Breonna Taylor. The concurrence of these events created a “syndemic”, where health issues and racism interacted to impact population health more than usual [66].

#### 2.2.5. Participant Sampling

Using a purposive sampling strategy, we partnered with Baltimore City Health Department’s *Baltimore for Healthy Babies* to identify and recruit postpartum women from three early home-visiting programs in Baltimore, Maryland. We asked staff (i.e., home visitors) of these three postpartum home-visiting programs to evaluate the study’s eligibility criteria and refer clients who could be potential participants. A purposive sampling strategy allowed us to leverage staff of home-visiting programs to most easily and accurately find and access potentially qualified participants.

Participant eligibility criteria were as follows: 18 years or older, self-identified as an African American or Black woman, not currently pregnant, within one year of most recent delivery (regardless of birth outcome), English as spoken language, and self-reported diagnosis with one or more of the following cardiometabolic conditions: Pre-pregnancy chronic hypertension, diabetes, or obesity at any point prior to the most recent pregnancy; or most recent pregnancy complicated by a hypertensive disorder of pregnancy or gestational diabetes. Eligible participants agreed to participate in exchange for a $25 gift card from either Walmart or Amazon, as per their preference. 

#### 2.2.6. Instrumentation

We used concepts from our adapted conceptual framework (Figure 2) to inform the development of our interview instrument. Drawing from principles of intersectionality-based structural inequity inherent to Critical Race Feminism, and considering behavioral factors for healthcare utilization, we created a series of open-ended questions about how care-seeking behaviors, illness experiences, ideas about wellness, and patient–provider relationships have been shaped by participants’ identity as Black women and mothers (Appendix B). Interview questions were designed to uncover elements of structural racism as delineated by Critical Race Feminism while simultaneously considering how these elements informed postpartum primary care utilization. Because we aimed to center and privilege the ideas and expertise of Black women, participants were also asked to provide recommendations for improving postpartum primary care for women similar to them. Our questionnaire did not include specific questions directly related to the tenets of Critical Race Feminism theory or predisposing, enabling, and need factors in the behavioral model. Instead, we designed our questions with the expectation that participants’ collective responses would reflect these concepts. Our goal was to elicit participants’ lived experiences that align with Critical Race Feminism and could be mapped onto healthcare utilization drivers, including needs, barriers, and facilitators. To ensure the appropriateness and effectiveness of our research questions, we sought input from a group of nurse home visitors and community-based doulas who possess valuable familiarity with the target population. Specifically, we solicited their feedback on various aspects, including the content, organization, and structure of the questions. Additionally, we assessed the ease of comprehension and whether the questions were perceived as intended. This feedback was obtained six months prior to conducting the initial interviews, allowing us sufficient time to refine and optimize our questions for clarity and relevance.

#### 2.2.7. Interviews

Following the informed consent and collection of brief demographic information, interviews were conducted by two investigators that had training in qualitative research methods and shared race and gender concordance with the participants. The study is led by a Black woman scholar, and all the interviews were conducted by Black women. This deliberate decision is rooted in underlying Black feminist ideals [47], which recognize the central role of Black women intellectuals in shaping Black feminist thought. Black women’s unique experiences provide them with a distinct perspective on Black womanhood and the conditions of their oppression. Their critical insights into these structures of oppression are invaluable for understanding and addressing the challenges faced by Black women. Furthermore, Black women intellectuals, both within and outside academic spaces, are more likely to remain committed to Black women’s struggles even in the face of overwhelming obstacles or diminishing rewards. It is important to note that the emphasis on Black women’s leadership in producing Black feminist thought does not exclude others from participating. However, it acknowledges that those who live the reality and directly experience these unique challenges are best positioned to define their own reality and shape the discourse surrounding it (see the conflict of interest statement for the positionality of the research team). The interviews occurred between May 2020 and June 2021 via telephone or video conference platform and were recorded and transcribed. Interview duration varied among participants, from 45 to 90 min.

#### 2.2.8. Analysis

The analysis process began with a focused reading of all interview transcripts to gain familiarity with the data. Data were then analyzed through the theoretical lens provided by our adapted conceptual framework critical [43,44,48,49,50,59]. We used ATLAS.ti Version 9.0.24 and ATLAS.ti Cloud version (ATLAS.ti Scientific Software Development, GmbH, Berlin, Germany) to facilitate coding and organization of the transcripts. Through an iterative process, our team developed a codebook. We first created codes a priori, guided by the adapted conceptual framework and research questions. Examples of these codes include “institutional racism-neighborhood environment” and “pre-established primary care provider”. Next, we added emerging codes from the data analysis, such as “patient-provider relationships”, “individuation”, and “pain management”. Our final codebook captured all relevant participant comments, delivering confidence that all data were captured for analysis. Six of the interview transcripts were randomly selected, and all six were independently coded by two investigators. Intercoder agreement, defined as the degree of consistency or agreement between multiple coders who independently code the same set of qualitative data, was assessed to ensure the reliability of the coding process. We used a process of simple agreement (i.e., the percentage of decisions that are agreements [67]. Once 85% intercoder agreement was achieved, the remaining twelve transcripts were coded by one of those investigators. We noted that saturation had been achieved (i.e., no new themes were emerging by the last interview) and concluded the data analysis process. Finally, the team conducted a comprehensive analysis of the coded data, guided by our theoretical frameworks and adapted conceptual framework. We identified emergent patterns and themes and synthesized the data to best address our research questions. To ensure clarity and alignment with our objectives, we organized our results based on how they provided insights into each of the research questions. Specifically, for RQ1, which explored the cultural, social, and historical contexts related to health, healthcare, wellness, and identity, we presented findings that specifically addressed these aspects. Similarly, for RQ2, which focused on investigating the needs, barriers, and facilitators associated with postpartum primary care, our results were organized to provide a comprehensive understanding of these factors.

## 3. Results

Twenty-seven home-visiting clients were referred by home-visiting staff members for screening. This figure represents the number of potentially eligible participants in the aggregate at the time of our study onset. Four were deemed ineligible for not meeting all criteria and five were unable to be reached. Eighteen participants comprised our final sample. The mean age of the sample of Black postpartum women (*n* = 18) was 31.7 years, and annual household income was <$15,000 for five people, $30,000–$49,000 for four people, and >$50,000 for four people (Table 1). Fourteen (of 18) had Medicaid as their primary insurance, and twelve reported having a PCP both prior to pregnancy and following delivery. Fourteen (of 18) participants reported having had a postpartum checkup with an obstetric provider and five had seen or scheduled follow-up with their primary care within one year after delivery. 

We identified six major themes (Figure 3). Themes 1–3 address RQ1: Cultural, historical, and social contexts that may influence postpartum primary care utilization. Themes 4–6 address RQ2: Barriers, facilitators, and needs related to postpartum primary care utilization. Quotes embedded below are shortened for brevity; unabbreviated quotations are available in Appendix A (RQ 1) and Appendix A (RQ2). Themes, subthemes, and representative quotes are shown below. To improve brevity and clarity, representative quotes have been abbreviated while maintaining the intended meaning. We have included the unabridged version in Appendix A. Participants are denoted with (P).

### 3.1. Theme 1: Enduring Influence of Structural Racism

#### (1A) “The Layers of Stress Never Stop”: Social Determinants of Health Influence Health and Care-Seeking

Several participants described challenges related to social determinants of health for themselves and their children in the context of being in majority Black neighborhoods. Structural factors, especially those related to transportation and housing stability, intersected with poverty and were a barrier to care-seeking. Sometimes these factors were direct barriers:

“*…I think about the trip… [like] ‘Dang, I got to catch this many buses, or I have to pay this much to get there?’ I’m just not going to go. So transportation is one of the biggest things for me for why I don’t go to [healthcare] appointments*”-P9. 

Neighborhood structural issues, including concerns about neighborhood safety, formed indirect barriers to health: “*If I walk outside there are drug dealers on both sides… [and] I’m nervous they will start shooting*”-P19. Conditions of subsidized housing impacted health and self-care decisions: “*[there are] mouse holes in my walls, holes in the bathroom ceiling, no access to the kitchen… and I have to stay in stuff like this because I can’t afford anything else*”-P6. 

Additionally, the pandemic increased caregiving responsibilities and financial strain while limiting access to social support: 

“*I can’t do hair as often as I was [before the pandemic] because you can’t be around as many people…so it messed up my money… and because my daughter’s doing virtual learning I got to help her with that. I also have another son…I can’t depend on anybody else for that…there’s a lot that I have to do on my own*”-P8.

Indirect barriers to healthcare seeking created competing demands, making it challenging to prioritize personal health. P6 describes the relentless cycle: 

“*It’s like the layers of stress never stop and then being stressed really can bring the lupus out. I can’t afford to have a lupus attack because I have no one to take care of my kids… And so that makes a difference in my health too because I don’t really have the time to take care of myself because I have so much to worry about as it it*”-P6. 

### 3.2. Theme 2: Personally Mediated Racism in Healthcare and Beyond

We identified three distinct forms of personally mediated racism that influenced healthcare interactions and overall well-being: (1) Gendered racism, (2) obstetric racism, and (3) vicarious racism.

#### 3.2.1. (2A) “They Judge Us before They Know Us”: Impact of Gendered Racism on the Health Care Encounter

Several participants experienced personally mediated discrimination based on gender, race, or socioeconomic position that made them feel judged by providers. As one participant noted: “*I think the way they treated me had something to do with me being Black and getting social services… they think because of where I am from and that I get social services… I don’t take care of myself*”-P8.

Participants also described experiencing negative stereotyping that led to disparate care: “[*providers] start off with the belief that we are unsupported, single moms… and [since] we are coming here unsupported, they think they can treat you less than the standard*”-P12. These experiences occurred frequently and repeatedly across the life course from childhood to adolescence and into adulthood.

#### 3.2.2. (2B) “They Treated Me like an Inmate”: Obstetric Racism in the Perinatal Period 

Several participants noted discrimination during pregnancy, labor, and delivery, which was distinct from the discrimination they had experienced at other points in their lives. The most commonly reported concern was having labor and postpartum pain dismissed or inadequately managed. Requests for pain management and medication were often met with assumptions by healthcare providers: “*…they thinking I wanted to get high off the medication*”-P4. Some participants noted that the pain had to be severely uncontrolled before action was taken: “*I had to cry for the head nurse to look at my chart and see I wasn’t lying about not being able to take ibuprofen… I think it had something to do with me being Black*”-P4.

Participants described harsh and infantilizing language, as well as the dismissal of safety concerns about potential adverse outcomes. One participant reported that when a safety concern was raised, a nurse told her, “*It’s not that bad. You’re not dying. You’re just being overdramatic.*”-P15. The same participant made connections between the dismissal of her concern and a subsequent adverse outcome: “*I kept saying the epidural doesn’t work. If somebody had listened, maybe I would not have been fighting when I got cut [during c-section] and lost all that blood.*”-P15.

These adverse experiences impacted participants’ physical and mental health, as well as their decision-making around family planning and subsequent hospital-based pregnancy care. As one participant concluded: 

“*All Black women should just have their babies at home instead of the hospital because something bad is going to happen [here]*”-P19.

#### 3.2.3. (2C) “It’s Scary Being a Black Mother to Black Kids”: Vicarious Racism and Hypervigilance

Several participants highlighted the stress that comes from mothering Black children in a racist society. This stress was in part due to “*Watching young Black men being killed for no reason*”-P16. Many participants described constantly worrying about police brutality toward Black people and neighborhood-level violence (markers of structural racism): 

“*That’s something that always sat in the back of my head, keeping [my son] safe, especially living somewhere like Baltimore*”-P16. 

Participants described worrying about their children’s exposure to racism (termed vicarious racism), which resulted in a state of hypervigilance. As one participant noted: “*You can’t let your guard down and let your kids be kids*”-P9. She (P9) noted that while this stress was a common experience, known to “*any Black woman, especially if raising boys*”-P9. 

While participants described the influence of various forms of personally mediated racism throughout their life course, the ordinariness of these conditions seemed to either (1) limit the identification of these experiences as belonging to or a product of historical or present-day racism or (2) lead to an acceptance of racism as a phenomenon that can be navigated but not necessarily changed. As one participant noted: 

“*… I mean, [racism] happens so often and you do get to the point where you don’t feel it, it doesn’t affect you. You’re numb towards it because it just happens so often.*”-P18

### 3.3. Theme 3: Sociocultural Beliefs about Health and Healthcare

#### 3.3.1. (3A) “Instilled in Me”: Cultural, Social, and Familial Ideals about Seeking Preventative Healthcare

Participants were influenced by family members’ attitudes about preventative care. Many described care-seeking behaviors as a household norm, commonly practiced by mothers: 

“*Mom was [a] very, go to the doctors [type] … we always went to the eye doctor, dentist, primary care, that’s something that was big in our house.*”-P4

Participants described how their observations of family members’ interactions with healthcare providers, as well as their own experiences during childhood and young adulthood, shaped their perceptions. They discussed witnessing and personally experiencing dismissive treatment from providers, acknowledging that these experiences were likely related to their race. One participant compared her and her mother’s experiences with a delayed medical diagnosis:

“*…When my mother would complain about a knot on her hand [they later found out was cancer], no one would do anything about it, …and when I was a child with a breathing problem they did not take seriously. [The doctors] would say I didn’t have a breathing problem. I think it had something to do with me being Black, and I think it had something to do with my mother being Black*”-P4.

#### 3.3.2. (3B) “[Just] Pray It away”: Mental Health Stigma and Utilization

Participants highlighted the stigma associated with mental health challenges “*In the Black community, you always want to go pray something away, no one wants to talk about their mental health*”-P7. Another participant noted the judgment that came with seeking professional mental health care: “*Mental health and things like that, they’re frowned upon so if you going to see a psychiatrist… they [the community] kind of look down on you*”-P14. 

The stigma associated with mental health within the Black community was also compounded by a lack of access to mental health services due to socioeconomic challenges: “*Insurance doesn’t want to pay, and if you on medical assistance it can be even more difficult*”-P7. There were also perceptions that the health system did not place equal value on mental health compared to physical health. As one participant poignantly noted: “*Mental health, I feel like it’s a need. I don’t understand why it’s not as available as cardiology*”-P7.

#### 3.3.3. (3C) “I Am Here, I Am a Person” Counterstereotypes and Faith Foster Resilience and Agency

The participants self-identified with the counter-stereotype that stood in contrast to the negative societal perceptions of Black women. Participants used words such as “*strong*”, and “*survivors*”, who “*do what they have to do*” and described remaining positive despite challenging circumstances, some leaning on faith to do so. They displayed a relentless attitude and determination to advocate for themselves and their families. 

For example, one participant described her quest to find the right provider: “*He wasn’t listening to me, so I changed doctors. There comes a time that you can’t let that stop you…if someone isn’t listening to you or you don’t like that doctor, you can’t just stop and say you’re not going to go get the situation handled. You got to keep going, and you got to find someone else who will listen*”-P18.

### 3.4. Theme 4: Barriers to Postpartum Care Transitions

Most participants conceptualized “postpartum care” solely as a health evaluation with their obstetric provider to ensure they were recuperating after pregnancy and labor. Some participants also brought up postpartum depression as an important part of the postpartum obstetric follow-up. However, none of the participants (with chronic illnesses) considered how a postpartum visit with their PCP might be important for long-term prevention and risk stratification after a medically complicated pregnancy. 

#### 3.4.1. (4A) “I Didn’t Think I Needed to See My Primary Care Doctor:” Limited Patient Knowledge about Postpartum Health Needs Following Cardiometabolic Complication of Pregnancy and Role of Primary Care Providers

Several participants believed that pregnancy complications (e.g., gestational diabetes and preeclampsia) were limited only to pregnancy and did not need continued attention following delivery. While most participants reported having prior knowledge or getting specific education about these cardiometabolic complications, none of the participants reported receiving education (before or during pregnancy) about the long-term cardiovascular risk and the need for primary care follow-up. Some participants viewed their obstetric providers as their PCPs and did not know about the different scopes of practice between obstetric providers vs. PCPs. This seemed to deter primary care follow-up: “*I haven’t seen my primary care doctor in a long time… because my GYN doctor did the things that needed to be done*”-P19.

#### 3.4.2. (4B) Lack of “Information Sharing” between Obstetric and Primary Care Providers

Some participants perceived ineffective transitions of care and communication between obstetric providers and PCPs throughout pregnancy and postpartum, including reports of a lack of collaborative mindset and turf battles. For example, participants reported that having different hospital systems for obstetric providers and PCPs hindered inter-provider communication. Several described needing to be the liaison and own advocate by sharing information between their obstetric providers and PCPs: “*I wasn’t sure if my OB was sharing information with [my primary care provider] during the pregnancy. So, I would just email my primary doctor every so often during the pregnancy just to let her know how I was doing*”-P16.

### 3.5. Theme 5: Facilitators of Postpartum Care Transitions: Patient-Provider Relationships

Participants identified characteristics of various patient–provider relationships before, during, and after pregnancy that influenced their perception of healthcare and providers.

#### 3.5.1. (5A) “Not just Another Chart”: Patient-Centered Care, Humanistic Care and Individuation Improved Experiences of Care

Many participants voiced that being treated like “*just a number*” consistently turned them off from a provider. They also shared counter-experiences of individuation (i.e., feeling known as a person), that helped to strengthen relationships and helped facilitate shared decision-making. For example, one participant shared: “*He [primary care provider] knows me. He remembers what we talked about last time… He makes me feel like he’s listening [and] yeah, I’m not just another chart… I feel comfortable coming to him saying, ‘No, let’s make another adjustment’. And he’s open to doing that. So, we work through it together.*”-P18.

Similarly, “*being treated like a human being*” was valued in the patient–provider relationship and was a stark contrast to the dehumanization reported during negative patient–provider encounters. As one participant noted: 

“*I like it when someone is attentive, and they’re able to talk to me like I’m a human, and not just like, ‘You’re just a number.’ My primary care provider would answer all of my questions, make sure everything was understood before I left [in contrast], I had a provider [during pregnancy], and I did not like her because she didn’t really take the time out to explain stuff or really got to know me or tell me what I should be expecting out of this…*”-P11

#### 3.5.2. (5B) “I Keep Him in the Loop”: Preconception Relationships Improve Care Continuity and Communication

Several participants had long-term, even multi-generational relationships with their PCPs. This continuity of care helped foster trust over time and appeared to be an enabling factor for care-seeking, and engagement. Of note, participants who were their own advocates bridging the communication gap between obstetric providers and PCPs had strong ties with their PCPs before delivery, and continued communication during pregnancy and postpartum.

“*I’ve been a patient of hers for, I’d probably say, maybe 8 or 10 years. I did discuss [plans to get pregnant] with her last time I saw her; I’d asked her when I got on the blood pressure medicine, was it safe to be on this specific medicine while I was pregnant and for breastfeeding.*”-P16

#### 3.5.3. (5C) “I Didn’t Have to Put on a Professional Face”: Racial and Gender Concordance Supports Open Communication

While some participants did not have a choice about having racial and gender concordant providers, several noted that they sought out Black women providers—that there was a “*comfort level with talking to a Black doctor because she looks like me*”. Racial and gender concordance also took away the need to change their behavior (e.g., speech) to ensure they were perceived positively.

### 3.6. Theme 6: Postpartum Health and Healthcare Needs

#### (6A) “Postpartum Is No Joke”: Lack of Preparation for Unique Mental Health and Social Support Needs That Emerge Postpartum

Many participants felt that there was less education and preparation for the postpartum period compared to pregnancy. In addition, mental health and particularly anxiety and depression were commonly described as requiring additional, ongoing attention. Importantly, isolation often compounded these feelings, and participants also described a need for peer support, beyond pregnancy. As one participant noted:

“*Even with young mothers, young Black mothers…I think if they had more groups where they could all sit around and realize that they’re not the only one that’s scared and stuff like that and exchange number.*”-P4.

The same participant noted that accessing support was easier during pregnancy and expressed the desire for a similar level of access to support in postpartum primary care. “*You get a little more attention when you’re pregnant because there’s so many things that can go wrong… they had groups every week… you talk with other mothers that have a baby around the same time as you, they have resources, like discounted car seats…I think if they had more groups and stuff like that…even in a setting where the primary care doctor probably [could] offer a group or something-- I think that’d be helpful.*”-P4.

## 4. Discussion

The goal of this study was to center the experiences of Black mothers to gain a better understanding of drivers of postpartum primary care utilization. Interestingly, but perhaps not surprisingly, given the fragmented state of postpartum/interpregnancy care, we found that most participants were not aware of their need for postpartum primary care. We also found that (1) racism (structural, interpersonal, and vicarious) is an omnipresent factor that affects overall emotional and physical wellbeing, poses a competing demand in prioritizing health, and can ultimately influence participation in health care. (2) Black women’s sociocultural experiences throughout the life course influences how they arrive at and engage with healthcare. (3) Pregnancy and childbirth are influential experiences that substantially impact postpartum health and may be overlooked learning opportunities for enhancing education and communication around cardiometabolic health. (4) Healing from complex or traumatic perinatal experiences occurs over a continuum and requires continued attention to both physical and mental health in the postpartum period. (5) Patient–provider relationships are critical to shaping the care experience: Those grounded in dignity, respect, and individuation improve the care experience as does continuity of care and patient–provider concordance, which can facilitate rapport building and enhance trust. 

The theoretical frameworks supporting this research provided sufficient latitude to probe the ways in which racism influences the daily lives, healthcare experiences, and healthcare utilization of Black postpartum mothers. To our knowledge, only one other study has qualitatively explored the receipt of preventative health in the perinatal period among Black women through a framework that examines racism’s function and intersectionality of gender, class, and race. In 2010, Hogan et al. published an ethnographic study of 19 Black women using intersectionality theory to describe how social and structural disadvantage shape health-related behaviors, risk perception, and vulnerability in the preconception and interconception period [38]. The authors found that structural racism intersecting with class, gender, and history created unique exposures for Black women. Similar to our findings, they concluded that the burden of compensating for social disadvantage takes its toll on one’s well-being, leaving little energy to prioritize health.

Our research shows that the usual social and structural barriers that limit Black women’s participation in healthcare are compounded and magnified in the stress-filled and dynamic (i.e., requiring constant change and transformation) postpartum period. However, the postpartum period may also allow for increased clarity on what works (facilitators) and what additional support is required (needs). 

With grounding provided by Critical Race Feminism and the revealed contextual factors clear, opportunities to address these identified drivers of postpartum primary care utilization emerge at systems, provider, and community levels. Below we describe these opportunities as illuminated by our research questions, and as implications for future action. 

### 4.1. Implications

#### 4.1.1. Contextual Factors and Structural Solutions

Participants in this study noted the way in which neighborhood violence, poverty, and police brutality (all forms of structural racism) take up mental real estate and pose a competing demand to prioritize health. Participants also noted other downstream factors such as not having access to childcare for other children, having unreliable or inconvenient transportation, and not having convenient clinic appointments (i.e., no group appointments for multiple family members, or no appointments after work hours) as barriers to postpartum care utilization. Importantly, healthcare systems have the power to address social determinants of health to support women in complying with recommended diagnostic or therapeutic plans. In Table 2, we highlight how our findings related to structural racism and social determinants of health could be specifically addressed through healthcare systems interventions. Some examples of these interventions include offering onsite childcare for clinical appointments or partnering with local organizations that offer childcare respite (e.g., The Family Tree [68]), making transportation assistance available for patients in need and seeking patient input on what type of transportation assistance would be most convenient, offering joint mommy–baby visits/family visits, and creating clinic visit slots that fall outside of work hours. However, these innovations address downstream consequences of structural racism and, ultimately, more upstream factors (i.e., poverty, educational opportunity, accountability for police brutality, etc.) must be addressed through interventions that “attempt to change the social, physical, economic, or political environments that may shape or constrain health behaviors and outcomes, altering the larger social context by which health disparities emerge and persist” [69]. Hospitals and healthcare systems have a role to play. They often employ a significant portion of the population in their catchment area, including trusted community members, and they have a significant influence on policymakers as they are uniquely positioned to create and proffer strategic plans that prioritize the elimination of health inequities [53]. Indeed, there is increasing recognition that hospitals and healthcare systems that are invested in equity, and especially those that predominantly serve historically marginalized populations, should view these issues as part of their fundamental responsibility.

In addition to findings of structural racism affecting health and healthcare utilization, our study adds to the growing literature describing how experiences of interpersonal racism and discrimination—related to gender, race, and social class—may lead Black women to avoid seeking preventive and reproductive health services [54,56,70,71]. Our findings extend the existing literature by illuminating obstetric racism and vicarious racism as unique forms of interpersonal racism and discrimination, beyond gendered racism, that harm Black mothers and may additionally impact their wellness and healthcare utilization.

Reduced interpersonal racism and bias in healthcare encounters require both individual introspection and reflection among health professionals, which can be supported by individual-level formal education [72,73,74], as well as an institutional commitment to culture change and systems of accountability [75] (Table 2). Improved measurement of patient experiences, including the use of surveys that address the unique experience of Black mothers, is a useful starting place [76]. More precise measurement can facilitate improved awareness about racism’s function in the lives of Black new mothers, and can support the provision of trauma-informed, patient-centered care; enhance healthcare systems’ accountability in addressing social determinants of health; and reduce disparities in utilization that contribute to persistent maternal and cardiometabolic inequities.

Despite the myriad challenges presented by structural and interpersonal racism, Black postpartum mothers rarely allowed racism to be the dominant narrative of their lives. Instead, they self-identified in positive ways (i.e., strong, a person who does not give up), and leaned into faith to persevere through challenges and show up for their children. This counter-stereotyping and counternarrative production lends itself to the development of resilience. However, this resilience may not be without consequence: In 2010, social epidemiologist Cheryl Woods-Giscombé described the strong Black woman role as it relates to stress and overall health in a conceptual framework, called the *superwoman schema* Woods-Giscombé postulated that the “*legacy of strength in the face of stress*” may contribute to health disparities among Black women [77]. From a cultural perspective, this self-identification as a Strong Black woman, or a superwoman, may explain some of the resistance to healthcare utilization in general and especially to formal mental health interventions, which we noted among participants [77].

While systemic barriers to accessing postpartum mental healthcare exist and should be addressed through system- and structural-level interventions (e.g., innovative mental healthcare delivery models (Table 2), healthcare providers and researchers need to consider how the *superwoman schema* might contribute to underutilization of mental health care among postpartum Black women. In addition, community dialogue that acknowledges the double-edged sword of the superwoman archetype may also give Black women the perspective necessary to reduce their own community’s stigma related to mental health treatment. 

**Table 2 ijerph-20-06283-t002:** The themes, subthemes, and opportunities related to Research Question 1.

Theme	Subtheme	Opportunities for Intervention at the following Levels: Provider (P), Community (C), System (S) ^+^
**Enduring Influence of Structural Racism**	“The Layers of Stress Never Stop”: Social Determinants of Health Influence Health and Care-Seeking.	As anchor institutions, hospitals, and healthcare systems should make investments in the community a strategic priority and allocate resources accordingly. The community needs assessment can be used to guide investment strategy. (S) Hospitals should invest in transportation support that (1) meets the needs of patients (e.g., transportation to accommodate those with disabilities, those with multiple children, etc.) and (2) does not create more barriers (e.g., bus tokens given, but multiple buses required to arrive at the destination, complicated or inefficient scheduling). (S) Hospitals should invest in childcare at the clinic or childcare respite programs for patients. (S) Hospitals should build clinical capacity (e.g., additional staff/locum providers or compensated late/early stay call system) to allow for flexible clinic hours. (S) Clinics should improve clinical efficiency through family/community-centered innovations such as group-based care, joint mother and baby visits, and improved referral to home-visiting programs. (S) Telemedicine should be an option available to patients based on individual preference and access to internet/digital technologies. (S) Hospitals should partner with community-based organizations to find services to address social determinants of health. Funding should be allocated to support these partnerships. (C, S) Providers should acknowledge the social determinants of health in their review of systems/medical documentation. (P)
**Personally Mediated Racism in Healthcare and Beyond**	“They Judge Us before They Know Us”: Impact of Gendered Racism on the Health Care Encounter.	Hospital systems should institute a racial discrimination patient safety reporting system. (S) Bias, antiracism, and upstander training should be required for all providers. (P, S) Training on tenets of patient-centered care, with an emphasis on acknowledging lived experiences and unique stressors/sources of trauma for Black mothers, (e.g., obstetric racism and vicarious racism) should be required for all providers. (P, S)
	“They Treated Me like an Inmate”: Obstetric Racism in the Perinatal Period. “It’s Scary Being a Black Mother to Black Kids”: Vicarious Racism and Hypervigilance.	Regulators should create billing codes to capture the impact of racism on health. (S) Hospitals should evaluate the role of racism on adverse outcomes and safety during maternal morbidity and mortality reviews [78]. (S) Hospital systems should utilize patient-reported experience measures unique to Black mothers, such as the PREM-OB scale [76]. (S) Providers should acknowledge the sociopolitical climate in medical documentation. (P) Providers should codify the impact of racism by using billing codes for mental health (i.e., acute, and chronic post-traumatic stress disorder). (P)
**Sociocultural Beliefs About Health and Healthcare**	“Instilled in Me”: Cultural, Social, and Familial Ideals about Healthcare.	Providers should actively inquire about cultural, social, and family ideals during history taking and integrate these factors into the process of shared decision-making. (P)
	“[Just] Pray It Away”: Mental Health Stigma and Utilization.	Clinics should reorganize to facilitate the co-location of services, such as psychiatry, obstetrics and gynecology, and primary care (e.g., internal medicine, family medicine, and pediatrics), in order to enhance overall efficiency and coordination of care (S) Clinic should hire mental health providers with perinatal health, infant-parent interactions/bonding, and healthy attachment expertise. (S) Destigmatize utilization of mental health resources- Black public figures and role models can aid in this. (C)
	“I am Here, I am a Person”: Counterstereotypes and Faith Foster Resilience and Agency.	Clinics should incorporate CenteringPregnancy [79] or alternative group visit models for women with similar pregnancy complications and gestational ages. (S) Hospitals and clinics should partner with faith-based organizations to provide additional support through specific groups for Black postpartum mothers. (S)

^+^ More than one code indicates opportunities present at multiple levels.

#### 4.1.2. Postpartum Primary CARE Utilization 

While reviews and consensus guidelines for postpartum primary care clinical management exist [9,80,81], there is a dearth of evidence about how postpartum care can better meet the multivariate social and structural needs that support healing and wellness [82]. *Bridging the Chasm between Pregnancy and Health over the Life Course: A National Agenda for Research and Action* (2021) was a Patient-Centered Outcomes Research Institute-funded stakeholder engagement process that provided a framework for postpartum transitions through systems innovation [16]. The authors described key strategies for remedying the siloed nature of women’s health care: The elimination of disrespectful and biased care, investing in communities and building capacity of a community-based organization, the transformation of care models (i.e., group models of care, team-based care), policy implementation that supports equity (i.e., Medicaid expansion), and centering women’s experiences and stories in data collection and research agenda setting. We similarly identified patient- and systems-level barriers (e.g., patient knowledge, cross-disciplinary information sharing, and care coordination) as well as facilitators (patient–provider relationships) and needs (mental health and social support needs) to begin to re-design systems. In Table 3, we highlight opportunities for intervention including educational and media campaigns focused on postpartum primary care concepts, establishing multidisciplinary postpartum care clinics [83,84,85], electronic medical record reminders [86], and templates to facilitate cross-disciplinary communication, enhanced medical education focus on the delivery of respectful care and cultural humility [87,88,89], increasing patient–provider concordance through diversity and inclusion medical pipeline efforts [90,91,92], improving engagement with community-based organizations to enhance social resource referrals for patients, and enhanced availability of postpartum support groups.

#### 4.1.3. Advancement of Theory and Praxis

Our research applies Critical Race Feminism theory to healthcare utilization drivers in the postpartum context, leveraging a valuable lens through which to better illuminate and understand the lived experiences of Black postpartum mothers. The underlying tenets of Critical Race Feminism, as illustrated in Figure 1, surfaced in our evaluation of the collective experiences of this population. Thus, our research expands the contexts for which Critical Race Feminism can help explain intersectional experiences.

In addition, due to the focus of our research on postpartum primary care utilization, our inquiry required a theoretical framework that could more specifically address healthcare utilization factors. While we chose the Gelberg–Andersen model for vulnerable populations as our healthcare utilization framework, our concurrent use of Critical Race Feminism helped us to identify where this model fell short: The explicit inclusion of racism as an underlying critical component of the experiences of vulnerable populations. The concept is implied in categories of social structure and sexual orientation for predisposing factors of vulnerable populations but is not explicitly included in the model. Furthermore, the ways in which these identity-related factors may intersect to create multiplicative disadvantage are not articulated in the current model. We believe our adaptation (Figure 2) more appropriately captures this for our population and propose that the historical and present-day functions of racism be considered in healthcare utilization frameworks for historically marginalized racial and ethnic groups. 

### 4.2. Limitations

Like all research, this study is not without its limitations. First, the participants selected to participate in this study had voluntarily enrolled in home visiting programs; accordingly, this sample may represent women who are more engaged with health care and related services at baseline, which could differ from the general population of Black women with cardiometabolic risk factors. There are likely additional barriers to postpartum primary care utilization among those who opt out for additional health care services; however, it may also be beneficial to learn from what motivates those who are more engaged with health care and related services. Second and relatedly, the participants in this study were generally well-resourced in terms of access to care and insurance coverage. Additionally, the income diversity within this group was limited, and we did not capture specific information regarding participants with higher incomes (>$50,000). As a result, the role of income in our analysis could not be adequately assessed. We recognize the need for more precise inclusion of socioeconomic indicators and a more diverse sample in terms of socioeconomic position, including access to resources for care. Third, as discovered through conducting this study, postpartum primary care is not a familiar concept to most women. Thus, we may have captured barriers and facilitators related to postpartum care in general and not specific to postpartum primary care as intended. Fourth, there are the usual issues with self-reported data (e.g., social desirability bias). We aimed to minimize the tendency to provide socially desirable responses by matching the interviewer in race and gender with the interviewee, but self-reported research always carries this inherent risk. Finally, we did not inquire about the nativity status of Black women in this study. There is substantial supporting literature that posits that the experiences and maternal health outcomes of Black immigrant women may differ from US-born Black women [95,96,97]. In future studies, we will include nativity status to better understand the diversity of the diasporic experiences of Black motherhood. 

## 5. Conclusions

To our knowledge, this is the first qualitative study exploring the concept of postpartum primary care among Black women with cardiometabolic risk factors. Consistent with critical race feminist theory, our study explicitly asked about gendered racism to contextualize barriers, facilitators, and needs related to postpartum primary care utilization and uncover areas for multilevel interventions.

Hogan et al. argue that “an improved understanding of the unique contextual experiences of Black women will arise only when all manifestations of racism are universally acknowledged and understood by public health practitioners” [38]. We agree with this assertion and emphasize that such acknowledgement and understanding can raise awareness of racism’s function in the lives of Black women. This, in turn, can support the provision of trauma-informed, patient-centered care; promote healthcare systems’ accountability in addressing social determinants of health; and reduce disparities in utilization that contribute to persistent maternal and cardiometabolic inequities However, awareness alone is not sufficient. Fidelity to principles of Critical Race Feminism demands action. The action must be grounded in the strengths of Black women, while simultaneously not burdening them with the task of creating improvement in spaces—where systems ought to shift.

## Figures and Tables

**Figure 1 ijerph-20-06283-f001:**
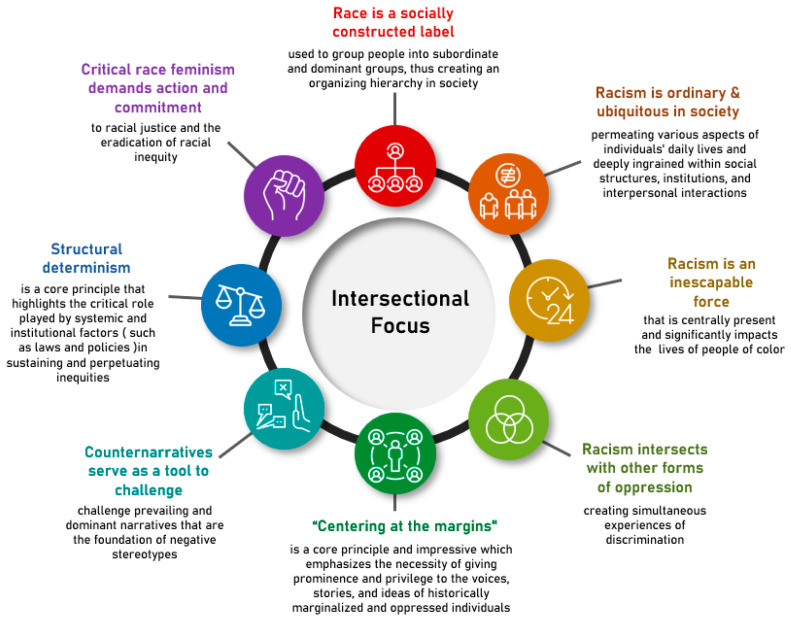
Critical Race Feminism theory: Critical Race Theory Core tenants + Critical Race Feminism distinction [43,44,48,49,50]. Critical Race Feminism: Critical Race Feminism is a theoretical framework that focuses on women of color who face multiple and intersecting forms of discrimination based on race, gender, and class (intersectional focus). It has several core tenets (outer surrounding circles). Critical Race Feminism is an outcropping of Critical Race Theory and similarly examines how race and racism operate within society. However Critical Race Feminism focuses specifically on the unique challenges and struggles confronted by women of color within the context of a system dominated by white male patriarchy and racist oppression.

**Figure 2 ijerph-20-06283-f002:**
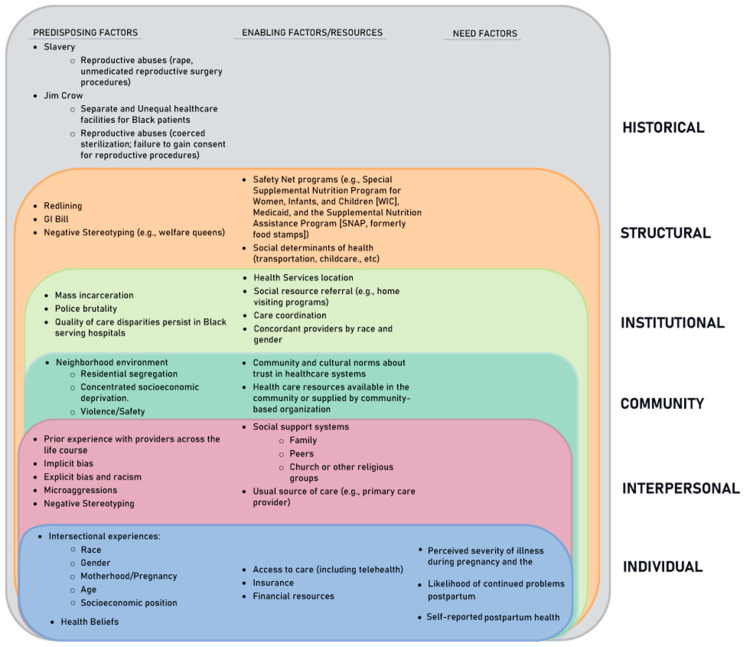
Conceptual Framework: Adapted Gelberg–Andersen Behavioral Model for Vulnerable Populations using Critical Race Feminism Theory [43,44,48,49,50,59].

**Figure 3 ijerph-20-06283-f003:**
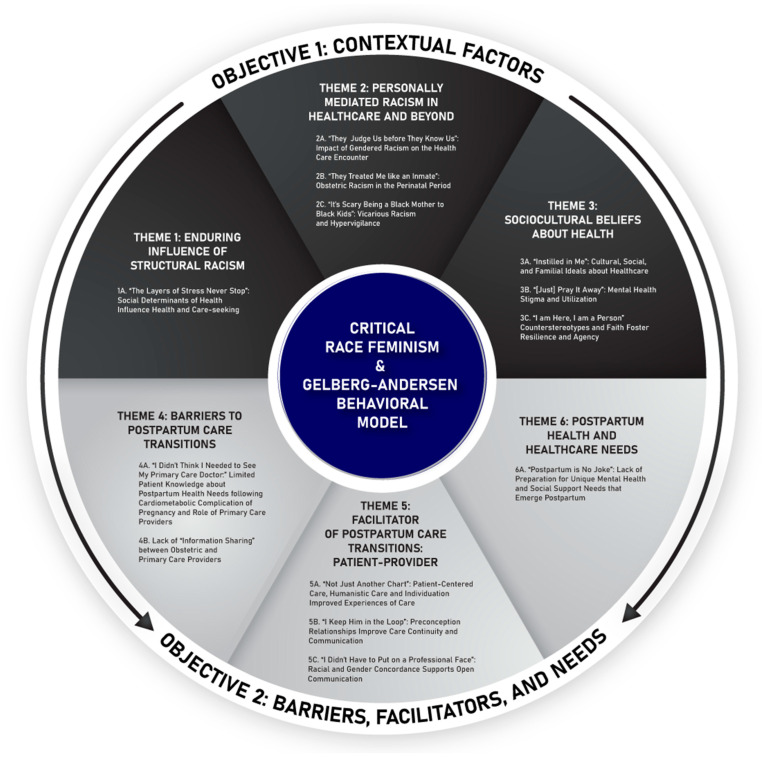
Themes and Subthemes informed by the Conceptual Model.

**Table 1 ijerph-20-06283-t001:** Characteristics of 18 postpartum Black mothers with cardiometabolic risk factors from three home visiting sites.

	N	(%)
**Social and Demographic**		
**Maternal age, years; range, (mean)**	18	23–43 (31.7)
**Yearly Income ($)**		
<15,000	5	27.8
15,000–29,000	3	16.7
30,000–49,000	4	22.2
>50,000	4	22.2
I don’t know	2	11.11
**Pre-pregnancy Chronic Medical Conditions**		
Diabetes	1	5.6
High blood pressure	6	33.3
Obesity	12	66.7
Sickle cell trait	3	16.7
Depression	9	50.0
Anxiety	10	55.6
Other	7	38.9
**Pregnancy-Related Comorbidities**		
**Gestational Diabetes**		
Yes	5	27.8
**Gestational High Blood Pressure**		
Yes	9	50.0
**Healthcare Utilization and Access**		
**Health Insurance Before Pregnancy**		
No Insurance	1	5.6
Medicaid	14	77.8
Private Health Insurance	3	16.7
**Health Insurance After Pregnancy**		
No Insurance	0	0.0
Medicaid	15	83.3
Private Health Insurance	2	11.1
Other Insurance	1	5.6

**Table 3 ijerph-20-06283-t003:** The themes, subthemes, and opportunities related to Research Question 2.

Theme	Subtheme	Opportunities for Intervention at the following Levels: Provider (P), Community (C), System (S) ^+^
**Barriers to Postpartum Care Transitions**	“I Didn’t Think I Needed to See My Primary Care Doctor:” Limited Patient Knowledge about Postpartum Health Needs following Cardiometabolic Complication of Pregnancy and Role of Primary Care Providers.	Create a distinct name for postpartum primary care visit (e.g., Primary Care After Baby [PCAB] visit) followed by a Media campaign to raise awareness. (S) Educate both obstetric providers and PCPs about the PCAB visit and encourage PCPs to discuss it with patients before pregnancy, and obstetric providers to discuss it with patients during pregnancy. (S) Establish a multi-disciplinary postpartum/interpregnancy clinic for patients with cardiometabolic complications of pregnancy (see Appendix A for example programs). (S) Improve obstetric provider knowledge on cardiometabolic complications of pregnancy and appropriate referral to PCPs. (P) Improve PCP knowledge about postpartum/interconception management of cardiometabolic complications of pregnancy [9]. (P) Establish disease specific postpartum group visits with PCPs who focus on women’s health. (C, P)
	Lack of “Information Sharing” between Obstetric and Primary Care Providers.	Collocate obstetric providers and PCPs and create a shared patient list of patients with cardiometabolic complications in pregnancy. (S) Implement a universal Electronic Medical Record to enable access, transparency, and communication between providers across hospital systems. (S) Create shortcut or dot phrases for discharge instructions for specific conditions. These should be shared among staff and entered into the electronic medical record as part of the discharge summary. (P, S) Hospital system should partner with postpartum community doulas, other community health workers, and home visitors to assist with postpartum care navigation postpartum. (C, S)
**Facilitator of Postpartum Care Transitions: Patient-Provider Relationships**	“Not Just Another Chart”: Patient-Centered Care, Humanistic Care and Individuation Improved Experiences of Care.	Educate providers on tenets of patient-centered care and shared decision making grounded in cultural humility and anti-racist principles [93]. (P) Educate providers on the principles of respectful care [87]. (P)
	“I Keep Him in the Loop”: Preconception Relationships Improve Care Continuity and Communication.	Increase continuity of care and opportunities for trust building by establishing patient cohorts. (S) Leverage pregnancy as an opportunity to connect mothers to primary care—this is a collaborative effort that requires communication and championing from obstetric providers and PCPs. (P)
	“I Didn’t Have to Put on a Professional Face”: Racial and Gender Concordance Supports Open Communication.	Improve diversity and inclusion of obstetric and primary care physician workforce, through systemic methods (e.g., affirmative action). (S)
**Postpartum Health and Healthcare Needs**	“Postpartum is No Joke”: Lack of Preparation for Unique Mental Health and Social Support Needs Emerge Postpartum.	Clinics should incorporate CenteringPregnancy [79] or alternative group visit models focused on Black postpartum mental health. (S) Connect mothers to postpartum support services: home visiting, lactation support, and postpartum doulas. Hospitals should invest in and finance these support services. (S) Improve cross-disciplinary services and care integration to effectively identify women during pregnancy and postpartum who require or may benefit from ongoing mental health support [94]. (S)

^+^ More than one code indicates opportunities present at multiple levels. Patients are intentionally excluded from this list of opportunities for interventions. This exclusion is based on a historical tendency to blame mothers and promote narratives of personal responsibility, which perpetuates inequity. Instead, by proposing the implementation of systemic and individual changes as providers and holders of privilege, we can create opportunities for patients to access and pursue better options.

## Data Availability

The data presented in this study are available in Appendix A.

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
