# Peer review of "Health Experiences of African American Mothers, Wellness in the Postpartum Period and Beyond (HEAL): A Qualitative Study Applying a Critical Race Feminist Theoretical Framework"

_ijerph, 2023, doi:10.3390/ijerph20136283_

Round 1
Reviewer 1 Report
This paper explores factors that influence postpartum primary care utilization for Black women in the United States. Informed by critical race feminist theory and Andersen’s model of healthcare utilization, the researchers aim to deconstruct popular narratives that reinforce negative stereotypes and contribute to the stigmatization of Black women while providing a deeper understanding of the drivers that influence postpartum primary care utilization. The findings from this study can inform the design of interventions that improve Black postpartum women's health.
The paper is well-structured and addresses an important gap in the extant literature on the experiences and primary utilization of Black postpartum women. Below are a few suggestions for improvement:
- The introduction does not adequately explain the role of racism in maternal mortality disparities. The role of racism in maternal mortality disparities is well-studied and should be discussed in the introduction.
- The authors should review the paper for consistency in the capitalization of the terms “Black” and “White”.
- Did the authors gather demographic data on women's ethnicity? Immigrant vs. non-immigrant women? There is some evidence that maternal/child health outcomes are better for some Black immigrant women subpopulations, depending on the length of residence in the United States and myriad complex factors. Such heterogeneity in such a small sample may obscure the unique experience of varied subpopulations and could be a limitation of the study.
Reviewer 2 Report
The goal of this study was to explore the cultural, social, and historical factors that affect 19 postpartum primary utilization among Black women with cardiometabolic risk factors and to 20 identify needs, barriers, and facilitators, that are associated with it.nThe topic is of significance because it addresses the drivers of postpartum primary care utilization for Black women, considering cultural, social, and historical contexts about health, healthcare, wellness, and identity. Additionally, the study looks at the needs, barriers, and facilitators of postpartum primary care for Black women. Prior studies have not generally not explicitly focused on Black women, or grounded in theoretical frameworks that help to interrogate the root causes of health inequities.
comments:
The study will be more powerful if additional groups or general population relations are used as controls to assess whether there is a significant difference between the general population and black women. White women have been referenced in the introduction. An option would be to contrast the findings from black women with white women to improve the study.
The themes indicated in the study could have data associated with them for easier reading and assist with answering the main question
Reviewer 3 Report
Thank you for the opportunity to review “Health Experiences of African American Mothers, Wellness in the Postpartum Period and Beyond (HEAL): A Qualitative Study Applying a Critical Race Feminist Theoretical Framework.” This was a well-designed and written manuscript. I found it so compelling and gained valuable insight by reviewing it.
I do not have any major concerns about this work. I consider all of my comments to be minor clarifying or wording edits.
Specific edits follow:
Figure 1: I don’t see a reference to Figure 1 until the Discussion. I’d like to hear more about where this Critical Race Theory Tenants figure comes from. You have some references, but I’m not sure whether it is an adaptation or directly from one of these resources. A few things that might make it clearer: 1) why is “women of color” in brackets in the center circle? 2) The different posts are in different tenses. So some say “race is…” but I’m not sure what the others mean. For example, “An acknowledgement of structural determinism”- does this mean that critical race feminism is an acknowledgement of…? And could each one then say critical race feminism is an acknowledgement that race/racism is…
Line 177: Does this statement refer to Baltimore specifically or within the US?
Line 195: Is this diagnosis within a certain timeframe? Could it have been during pregnancy up to a year ago, or is it a current diagnosis?
Line 203 and 246: I’m not familiar with the term saturation. Could you say more about how saturation is determined, and how that influenced your participant sampling? This seems to be how you determined sample size, is that correct?
Line 222: I’d like to hear more here about how the interview questions were developed. Did you use existing questions from other studies? You say that you solicited feedback, could you give more details about this- what was this timeframe? How were the questions designed?
Line 227: Race and gender concordance seems important, and you mention this again later. Maybe emphasizing here why that was in the study design here would emphasize this importance.
Line 237: I’d like hear a little more about your data analysis process. For example, how many codes did you have all together? How many a prior vs added once you were analyzing data? Why were there two investigators reaching agreement? Were these investigators chosen because of their expertise in this type of data coding?
Line 244: How was intercoder agreement calculated?
Line 253: Do you have an upper ceiling for $50K?
Figure 3: I love this graphical representation of your findings! Really great. These themes that you identify are so poignant. I really liked how you included direct quotes in your results, it made the data and themes really come alive and I found it heartbreaking but informative to hear from these interviewees in their own words and about their own experiences.
Line 463: The first sentence is a fragment.
I found Table 2 and Table 3 hard to read in this format. Since you have coding (e.g., S, C), perhaps the separate columns aren’t necessary. Or maybe the columns are more like checkmarks for which level, so that the text could take up more space. I don’t know how this will look in print form, so this might be more of a decision for editors. But as it stand, it’s a little hard to follow.
Table 2 under theme “enduring influence of structural racism”, last bullet point: Individual is misspelled.
Table 2 under theme “personally mediated racism in healthcare and beyond”: it was hard to tell what opportunities related to which subthemes. The alignment of the columns was strange.
Table 3: You’re missing a title here, so I was a little confused about how this table differs from table 2. I think this is more related to your second research question?
Line 612: “While we chose…” is a fragmented sentence.
Round 2
Reviewer 2 Report
The reviewer comments have been adequately addressed.